# Unpredicted Aberrant Splicing Products Identified in Postmortem Sudden Cardiac Death Samples

**DOI:** 10.3390/ijms232012640

**Published:** 2022-10-20

**Authors:** Monica Coll, Anna Fernandez-Falgueras, Anna Iglesias, Bernat del Olmo, Laia Nogue-Navarro, Adria Simon, Alexandra Perez Serra, Marta Puigmule, Laura Lopez, Ferran Pico, Monica Corona, Marta Vallverdu-Prats, Coloma Tiron, Oscar Campuzano, Josep Castella, Ramon Brugada, Mireia Alcalde

**Affiliations:** 1Cardiovascular Genetics Center, Institut d’Investigació Biomèdica de Girona (IdIBGi), 17190 Salt, Spain; 2Cardiology Service, Hospital Dr. Josep Trueta, University of Girona, 17007 Girona, Spain; 3Faculty of Medicine, University of Vic-Central University of Catalonia (UVic-UCC), Can Baumann, 08500 Vic, Spain; 4Centro Investigación Biomédica en Red de Enfermedades Cardiovasculares (CIBERCV), 28029 Madrid, Spain; 5Medical Science Department, School of Medicine, University of Girona, 17004 Girona, Spain; 6Forensic Pathology Service, Institut de Medicina Legal i Ciències Forenses de Catalunya (IMLCFC), 08075 Barcelona, Spain

**Keywords:** sudden death, intronic variants, splicing, genetics

## Abstract

Molecular screening for pathogenic mutations in sudden cardiac death (SCD)-related genes is common practice for SCD cases. However, test results may lead to uncertainty because of the identification of variants of unknown significance (VUS) occurring in up to 70% of total identified variants due to a lack of experimental studies. Genetic variants affecting potential splice site variants are among the most difficult to interpret. The aim of this study was to examine rare intronic variants identified in the exonic flanking sequence to meet two main objectives: first, to validate that canonical intronic variants produce aberrant splicing; second, to determine whether rare intronic variants predicted as VUS may affect the splicing product. To achieve these objectives, 28 heart samples of cases of SCD carrying rare intronic variants were studied. Samples were analyzed using 85 SCD genes in custom panel sequencing. Our results showed that rare intronic variants affecting the most canonical splice sites displayed in 100% of cases that they would affect the splicing product, possibly causing aberrant isoforms. However, 25% of these cases (1/4) showed normal splicing, contradicting the in silico results. On the contrary, in silico results predicted an effect in 0% of cases, and experimental results showed >20% (3/14) unpredicted aberrant splicing. Thus, deep intron variants are likely predicted to not have an effect, which, based on our results, might be an underestimation of their effect and, therefore, of their pathogenicity classification and family members’ follow-up.

## 1. Introduction

Unexpected sudden deaths remain a significant challenge to healthcare systems and forensic examinations worldwide. Sudden deaths can be classified as sudden cardiac death (SCD) when SD has a cardiac primary origin or as SD from non-cardiac origin. In SCD cases, molecular screening for pathogenic mutations in SCD-related genes is common practice [1].

Previous studies have been performed to genetically analyze SCD cohort in Catalonia during the last 10 years [2,3]. Those studies showed that less than 3% of total cases carried exonic pathogenic variants (P) and 26% likely pathogenic (LP) variants. Thus, test results may lead to uncertainty because of the identification of variants of unknown significance (VUS) in more than 70% of cases in the concurrent absence of clearly pathogenic mutations [1,2]. The presence of VUS variants is an uninformative genetic result in terms of clinical management, and genetic and clinical professionals cannot make informed clinical decisions based on such a classification. Current guidelines for the interpretation of rare variants include more items to be considered than those of 10 years ago [3]. These changes favor accuracy in classification but also increase stringency; thus, a lack of data for some of these items sometimes leads to ambiguous classification. Therefore, there is a certain number of variants classified currently as VUS that might confer a real risk in SCD. Thus, the situation hampers identification of at-risk family members.

Genetic variants affecting potential splice site variants are among the most difficult to interpret. Mutations which affect splicing are significant contributors to rare disease but are frequently overlooked by diagnostic sequencing pipelines. Splicing consensus regions are defined between positions 4 to 1 for acceptor site and from +1 to +6 for donor site. Thus, potential splice sites can be divided into three groups: canonical acceptor site variants (4 to 1), canonical donor site in position (+1 to +6), and deep intron variants (within the intron but outside the consensus sequence). However, the major consensus nucleotides at the U2 spliceosome donor and acceptor sites are GT and AG, which are known as the most classical canonical positions [4]. In terms of disease-causing, these classical canonical mutations at the 5′ splice site (+1 or 2) and 3′ splice site (−1 or 2) are already known and predicted to cause exon skipping or intron inclusion [4]. In silico tools for predicting splicing defects of canonical splice positions have a key role in assessing the impact of variants of uncertain significance. Several in silico software tools have been developed to predict splicing effects as they relate to gain or loss of splice sites at the exonic or intronic level. In terms of pathogenicity assessment and variant classification following ACMG criteria, canonical positions are also differentiated between +1/2 or −1/2 and beyond that [3].

At least 20% of SCD are due to rare hereditary cardiac diseases, and the genetic evaluation relies on rare coding and intronic flanking regions [1]. However, in terms of interpretation and variant classification, intronic variants usually rely on in silico predictions because of the lack of experimental studies. The 5′ and 3′ canonical (+/−1 or 2) variants are the most well understood, and several studies reported that they lead to either exon skipping or intron inclusion [5]. However, there is less of a consensus regarding the effect of intronic variants beyond +/−1 or 2 and deep intronic variants which may be mutation-specific. For this reason, these intronic variants are even more challenging to comprehend, especially when often they are found to be novel variants, and consequently most of them remain classified as VUS.

The aim of this study was to examine rare intronic variants identified in the exonic flanking sequence to meet two main objectives: first, to validate that canonical intronic variants produce aberrant splicing; second, to determine whether rare intronic variants predicted as VUS may affect the splicing product. To achieve these objectives, 28 heart samples of cases of SCD carrying rare intronic variants were studied.

## 2. Results

### 2.1. Characteristics of Population

The cohort of this study was selected from the total available cases of SCD that occurred in Catalonia in the past 10 years in the context of the MOSCAT project (Materials and Methods, Section 4). A total of 28 cases of SCD carried potential splicing variants in SCD-related genes. The mean age was 33.4 years old with a range of 18–49 years old and 60% prevalence of men. All 28 cases carried potential splicing variants in one of the studied SCD-related genes. From these cases, only two of them (6.67%) carried another missense variation classified as LP which may explain the SCD event (Case #12 *SCN5A*_c.1862del and Case #21 *TNNT2*_c.832C>T). The remaining 26 cases only carried VUS (Table 1).

### 2.2. Distribution of Variants

A total of 28 intronic variants were identified within the +/−25 pb in SCD-related genes in a forensic cohort of SCD. From these variants, 50% of them affected the canonical regions of the splice site while 17.85% were canonical acceptor site variants and 32.1% were consensus donor variants showing a higher prevalence of donor variants. Moreover, from these canonical splice variants, 14.28% affected the positions +*/*−1 and 2. The remaining 50% were located outside the canonical splice regions and were considered deep intron variants (Figure 1, Table 2).

### 2.3. Splicing Product Sequencing

#### 2.3.1. Canonical Site Variants

This study identified 14 intronic variants in the canonical sequence: 5 in the acceptor site and in the donor site. However, only 4 of them were located in the position +/−1 and 2, while the remaining 10 were located between positions −3 to −4 (acceptor site) and +3 to +6 (donor site). These types of variants are considered the most classical mutations in the canonical sites [4] (Table 3).

#### 2.3.2. The Most Classical Canonical Variants: In Silico Predicted Effect

The most classical mutations in the canonical acceptor sites are considered 1 to 2, affecting the NYAG/G sequence [4]. In our study, only two 3′ splice site variants were identified among SCD cases MYH6_c.2051-2del and CACNA2D1_c.659-2A>T. In silico predictors determined a high effect on the splicing product in both cases (Table 2 and Appendix A). However, experimental data showed very different results, with CACNA2D1_c.659-2A>T (Appendix A) presenting no aberrant splicing and MYH6_c.2051-2del showing exon skipping (Figure 2) leading to a premature stop at p.(Gly777Phefs*8). Moreover, only the aberrant isoform of MYH6 was identified in this patient, while the MYH6 wild type (including exon 18) was undetectable (Figure 2).

The most classical mutations in the canonical donor sites are considered +1 to +2, affecting CAG/GUAAGU [4]. In our study, two 5′ splice site variants within position +1 were identified among SCD cases: MYPN_c.2703+1G>T and TGFBR3_c.568+1G>A. In silico predictors determined in both cases as high in splicing (Table 2 and Appendix A). In both cases, a donor loss was predicted, which would lead to exon skipping.

Experimental results of cDNA sequencing confirmed aberrant splicing product in both cases. MYPN_c.2703+1G>T exon 12 skipping and TGFBR3_c.568+1G>A showed exon 6 skipping in heterozygosis, as predicted (Figure 2). In both cases, exon skipping lead to a premature termination codon (PTC); MYPN p.(Gly777Phefs*38) and TGFBR3 p.(Pro297Hisfs*18).

#### 2.3.3. Canonical Variants: Unpredicted Effect

In the canonical sequence, beyond positions +/−1 and 2, we identified 10 other mutations: 3 in the acceptor site and 10 in the donor site. All of them were predicted to have no effect on the splicing product by in silico platforms (Table 2 and Appendix A).

The three mutations identified in the acceptor site sequence in positions 3 and 4 were *MYH6*_c.3343-3C>A, *TNNT2*_c.690-4G>T and *FLNC*_c.5843-4C>T. In all cases, the results of the splicing product sequencing confirmed no aberrant forms supporting the in silico predictions.

Seven additional mutations were identified between positions +3 and +6 affecting the consensus sequence on the donor site (CAG/GUAAGU). Interestingly, from these seven variants, one variant *TTN* c.3224+5A>G produced unpredicted aberrant splicing product excluding exon 161 (Figure 3), which produces an in-frame deletion in the translated protein p.(Pro10735_Val10762del).

#### 2.3.4. Deep Intron Variants: Unpredicted Effect

A total of 14 deep intron variants were identified among SCD cases (Figure 1). In silico predictors determined a null or low effect on the splicing product in 13 of them (92.85%) (Table 2 and Appendix A) while predicting a moderate effect on variant AKAP9 c.95795C>G.

Experimental results, however, revealed that *AKAP9* c.95795C>G produced normal splicing product contrary to the in silico predictor results. For the remaining 13 variants predicted to have no effect on the splicing product, 11 were confirmed to produce a normal splicing product (84.61%). Among these variants, a particular case was found. *ANK2*_c.3224+7G>A was located in the 2728 intron. Although exon 28 was reported to be expressed in the heart, both the carrier and noncarrier control the same splicing product with the exclusion of exon 28 (Appendix A), supporting the lack of effect on the wild-type isoform.

However, two variants showed unexpected aberrant splicing product. Firstly, *TRDN*_c.1322-10T>A showed exon 21 skipping (Figure 2), producing an in-frame deletion in the translated protein p.(Met500_Gln515del), and secondly, *TTN* c.301796_301795insG showed complete intron retention (intron 142) with the inclusion of an in-frame PTC (Figure 2), i.e., c.30262_30263insN[207], leading to p.(Pro10086_Lys10087ins28*).

### 2.4. Correlation between In Silico Predictors and the Splicing Product

For the most canonical variants (+/−1 and 2), in silico programs predicted an effect in 100% of cases (4/4). On the contrary, none of the canonical site variants beyond +/−1 and 2 positions were predicted to affect splicing product (0/10). Experimental data showed aberrant product in 75% of the most canonical variants (3/4) while none of the remaining canonical site variants produced aberrant splicing product, supporting the in silico prediction. On the other hand, for deep intron variants, three aberrant splicing products were identified (3/14). All of them were unpredicted by the in silico programs. Thus, in this study, the in silico predictions for deep intron variants showed 0% sensibility and 83.3% specificity. Taking all data together, the overall accuracy of in silico predictors was determined to be 89.3% with a lower value for deep intron variants (78.6%) (Figure 4, Appendix A).

### 2.5. Aberrant Splicing Variants and SCD Events

#### 2.5.1. The Most Classical Canonical Splice Site Variants: Predicted Effect

Firstly, c.2051-2del variant in *MYH6* gene, which encodes for the alpha heavy chain subunit of cardiac myosin, showed aberrant splicing consistent with the predicted donor loss in the splicing consensus region. This aberrant splicing product led to a PTC which would presumably trigger the nonsense-mediated decay (NMD) machinery resulting in the degradation of the allele. Thus, this variant would be expected to cause MYH6 haploinsufficiency, but our results also showed undetectable wild type cDNA, suggesting a possible unexpected phenomenon of monoallelic expression.

Secondly, variant *TGFBR3* c.568+1G>A, which encodes for transforming growth factor beta receptor 3 which has not been defined as definitive in any SCD-related disease, was proved to lead to a PTC triggering NMD. Lastly, our study identified *MYPN* c.2703+1G>T in a case of SCD. This variant was predicted to cause donor loss in the splicing consensus regions which would lead to exon 12 skipping as was confirmed by our sequencing results, leading to a PTC p.(Gly777Phefs*38). Therefore, this variant would cause haploinsufficiency for MYPN. In this case, the circumstances of death are compatible with an SCD event (Table 1).

#### 2.5.2. Deep Intronic Variants: Unpredicted Effect

Deep intron variant *TRDN* c.1322-10T>A was found in a case of a patient who suffered SUD while sleeping. Molecular results showed abnormal splicing product excluding exon 21 by exon skipping. This case did not carry any other rare variant in SCD-related genes; however, the circumstances of death are compatible with an SCD event, possibly arrhythmogenic death.

TTN c.3224+5A>G was found to produce exon skipping, leading to an in-frame deletion in the translated protein p.(Pro10735_Val10762del). The carrier suffered a sudden cardiac death with no structural alterations being defined in the autopsy as a death of arrhythmogenic origin. Furthermore, this case also carried a likely pathogenic variant in SCN5A leading to a PTC.

Finally, *TTN*_c.301796_30179-5insG was found to produce a PTC p.(Pro10086_Lys10087ins28*). In this case, the patient was a 24-year-old female who suffered an SCD event with a possible toxicological status and did not carry any other LP variation. No structural phenotype was reported in the autopsy.

## 3. Discussion

In sudden death cases, molecular screening for pathogenic mutations in SCD-related genes is common practice. However, genetic test results are sometimes challenging to interpret due to uncertainty because of the identification of VUS in the concurrent absence of clearly pathogenic mutations. In this sense, intronic variants are very challenging to interpret and very frequently classified as VUS in the absence of experimental data.

To tackle this challenge, the aim of this study was to elucidate the functional effect of potential splice site variants in 28 cases of sudden cardiac death and evaluate whether there are discrepancies between computational predictions and experimental evidence.

### 3.1. Distribution of Variants and In Silico Prediction Accuracy

The results in our cohort show that consensus donor splice site variants are more prevalent than acceptor splice site variants. Our study presents a ratio of 1.8:1 in concordance with previous knowledge in other fields and based on larger previous studies describing a ratio of 1.5:1 [6].

Our results show 86.95% of the genetic variants predicted not to affect splice sites do not show any aberrant splicing form of mRNA, supporting the in silico prediction. On the other hand, only 60% of genetic variants predicted to highly affect splice sites have been confirmed experimentally to show an aberrant splicing product. Additionally, unpredicted aberrant splicing products were identified in around 10% of the total variants, suggesting that in silico predictions may sometimes underestimate effects of intronic variants, especially those variants not affecting the most classical canonical splicing site (1/2 and +1/2). All unexpected aberrant splicing products have been found either in deep intron variants or canonical variants beyond positions +1/2 or 1/2.

Interestingly, in our study, canonical splicing sites show much higher accuracy, sensitivity and specificity than deep intron variants. Our results set sensibility of canonical variants at 75% and specificity at 90%. In concordance with previous publications, splice site prediction tools have higher sensitivity relative to specificity in predicting splice site abnormalities [7,8,9].

Importantly, our results also suggest that intronic variants affecting the most classical canonical site (+/−1 and 2) are mostly predicted to have an effect (100%), while variants affecting canonical site from −3 to 4 and +3 to +6 are generally predicted not to affect splicing product (0%). In our study, experimental data showed that the classical canonical site (+/−1 and 2) led to aberrant splicing in 75% of cases, overestimating their effect, while canonical variants beyond +/−1 and 2 may be underestimated.

In this sense, deep intron variants showed no sensibility in our study; none of the abnormal splicing products detected were predicted by in silico tools while the only one predicted to have a moderate effect revealed a normal splicing product. These results suggest that deep intron variant predictions have very low sensitivity which might lead to an underestimation of the effect of these kinds of variants when classifying their pathogenicity. Thus, experimental results of splicing product sequencing showed that there is more complexity in the interpretation of rare intronic variants.

### 3.2. Genetic Causality of Splice Site Variant with Aberrant Product

Our study showed six aberrant splice products, three with unpredicted effects. In terms of their possible causality, taking the experimental results, autopsy details and previous literature together, we believe that *TRDN* c.1322-10T>A, *MYPN* c.2703+1G>T and *TTN*_c.301796_30179-5insG should at least be taken into account in further cases while the remaining three (*MYH6* c.33433C>A, *TGFBR3* c.568+1G>A and *TTN* c.3224+5A>G) did not show sufficient evidence to be considered the potential genetic cause.

*MYPN* c.2703+1G>T leads to a PTC p.(Gly777Phefs*38), and therefore this variant would cause haploinsufficiency for MYPN. *MYPN* has been described to be associated with both hypertrophic and dilated cardiomyopathy (DCM) [10], although their association is still considered to be limited. Thus, we suggest that this variant should at least not be discarded in further suspected SCD cases. Second, *TRDN* c.1322-10T>A located in *TRDN* encodes Triadin, a protein important for calcium release regulation from the sarcoplasmic reticulum. Genetic variants in *TRDN* have been widely reported to be the cause of the CPVT phenotype [10]. Further experimental studies should be performed to investigate and clarify their functional effect of losing exon 21 in the normal activity of TRDN. However, our results suggest that this variant should not be ignored as a potential cause of SCD events. Finally, *TTN*_c.301796_301795insG was found to produce a PTC p.(Pro10086_Lys10087ins28*). Loss of function variants in *TTN* has been considered to be pathogenic in the context of DCM [10]. This case was a 24-year-old female patient who suffered an SCD event and did not carry any other LP variation. It is not unusual for an SCD event to occur in early stages before a structural alteration is developed [11]. Thus, we strongly believe that our results should be taken into account when interpreting this variant in the future since it is plausible that *TTN*_c.301796_30179-5insG may be related to SCD.

On the other hand, MYH6 has limited evidence to support this gene disease relationship [10]. In addition, although *TGFBR3* c.568+1G>A would lead to a PTC triggering NMD, the circumstances of death showing possible myocardial infarction in the autopsy and this gene has been associated with Marfan syndrome in a single publication, though there is limited evidence to support this gene disease association [12]. Finally, *TTN* c.3224+5A>G was found to produce exon skipping, leading to an in-frame deletion in the translated protein. However, in this case the patient also carried a likely pathogenic variant in *SCN5A* leading to a PTC, which may be a stronger candidate as the causal mutation. The contribution of *TTN* c.3224+5A>G in the SCD event remains unknown, as well as the effect of this variant when identified solely.

Taking all of the above together, our data showed that in silico predictors alone are insufficiently reliable to determine the effect of splicing variants. Our study revealed some unexpected aberrant splicing products. Cardiac samples from SCD cases are very difficult to obtain and highly valuable to functionally evaluating genetic variants. We strongly suggest that, when possible, experimental data should be assessed as they are crucial to elucidating the role of intronic variants, and this could make the difference to properly evaluating the risk. Additionally, correlation with forensic information is key to the proper interpretation of genetic variants in terms of SCD outcome.

### 3.3. Study Limitations

This study has some limitations that should be mentioned. Firstly, the cohort size was limited in extrapolating the accuracy of in silico predictors. It is very challenging to find SCD cases carrying rare intronic variants with heart samples available to study the splicing products, and therefore experimental data may be valuable for future variant interpretations. Secondly, the distribution of variants among groups is heterogeneous because of the stochastic nature of genetic variant appearance. Third, the Sanger sequencing technique does not have the capability to detect sequences representing less than 5% of the total products. Thus, we cannot discard that all variants showing only the normal splicing isoform might be causing an aberrant isoform with less than 5% of relative expression.

## 4. Materials and Methods

### 4.1. Study Population

The present study was designed as a part of the MOSCAT project which encompasses the genetic study of all of sudden death (SD) suffered by individuals younger than 50 years old occurring in Catalonia (Spain) from 2012 to 2020. This study included the collection of heart samples of all cases of SD with a mandatory judicial autopsy occurring in Catalonia. The study was approved by the ethical committee of Hospital Universitari Dr. Josep Trueta de Girona (Spain) and conformed to the ethical guidelines of the Declaration of Helsinki 2008.

### 4.2. Forensic Analysis

Forensic autopsies were performed to determine the cause of death. This forensic autopsy included an exhaustive histologic and macroscopic cardiac exploration including histological tissue analyses, as well as toxicological investigation. In most cases, different parts of the heart were collected and frozen for future functional and molecular studies.

### 4.3. Molecular Genetic Analyses

Genetic analysis was performed in all individuals fulfilling inclusion criteria according to the flow chart (Figure 5) which included all individuals from 18 to 50 years old who suffered SCD and during the medico-legal autopsy were classified with a cause of death of suspected SCD. We excluded those cases where non-cardiac cause of death was identified and in those for whom the cardiac examination revealed cardiac ischemia as a cause of death. Genomic DNA was extracted from postmortem whole blood with the automated Chemagic MSM I (PerkinElmer, Waltham, MA, USA) instrument. Spectrophotometric measurements were performed to assess quality ratios of absorbance (260/280:260/230, minimum of 1.8:2.2). DNA concentration was determined using a Qubit fluorimeter (Thermo Fisher Scientific, Waltham, MA, USA). Immediately after DNA extraction, 3 μg was fragmented by sonication using the Bioruptor (Diagenode, Seraing, Belgium), and library preparation was performed according to manufacturer’s instructions of SureSelect XT (Agilent Technologies, Santa Clara, CA, USA). A custom resequencing panel which interrogates 85 related genes with SCD was used for the capture approach technology (Appendix A). Indexed libraries were entered in the sequencing path, and pooled captures were sequenced on an Illumina MiSeq instrument (San Diego, CA, USA) using 2 × 76 base pairs read length. DNA reads were mapped to an annotated reference sequence and determined the extent of variation from the sequence reference.

### 4.4. Variant Selection and Classification

All variants with a minor allele frequency (MAF) < 0.1% in PopMax Filtered in gnomAD were selected to be confirmed with the conventional Sanger technique. From filtered variants confirmed by Sanger sequencing, we selected all cases carrying intronic variants within the flanking region covered by the custom NGS panel (within +/−25 pb). Maxscore and SpliceA software programs were used to predict splicing effects. In addition, the Human Gene Mutation Database (HGMD) and PubMed were used to ascertain previously reported pathogenic mutations. Finally, all the variants were classified as pathogenic (P), likely pathogenic (LP) or variants of unknown significance (VUS) according to the standards and guidelines for the interpretation of sequence variants from the American College of Medical Genetics and Genomics and the Association for Molecular Pathology. All variants are described according to the last version 20.02 of HGVS released 1 May 2020.

### 4.5. RNA Extraction

Cardiac samples were stored at −80 °C from their initial collection until their processing (January–March 2022), with no prior freeze–thaw cycles. Total RNA was isolated from left ventricle heart tissue. A total of 25mg of frozen tissue was processed per sample. Tissues were disrupted and simultaneously homogenized by agitation in the presence of beads and lysis buffer in Tissuelyser LT (Qiagen, Hilden, Germany) for 2 min at 30 Hz. RNA was extracted using the RNeasy Mini Kit (74106, Qiagen) according to the manufacturer’s instructions. This protocol includes incubation of Proteinase K at 55 C for 10 min. All 28 cases suffered SCD in the period 2016–2021. Prior reverse transcription reaction was performed with an additional step of DNase I treatment and with gDNA Wipeout buffer. Reverse transcription reactions of RNA were performed using the QuantiTect Reverse Transcription Kit (205313, Qiagen). For each 20 µL reverse transcription reaction, 1 µg of total RNA was mixed with 4 µL of RT buffer, 1 µL of primer mix and 1 µL of Reverse Transcriptase in nuclease-free water. The cDNA concentration and 260/280 ratio were checked using a NanoDrop1000 spectrophotometer (Thermo Fisher Scientific, Waltham, MA, USA).

### 4.6. cDNA Sequencing

cDNA was analyzed using Sanger sequencing. Firstly, polymerase chain reaction (PCR) was used to amplify the interest regions. Primers were designed in coding regions of the nearby exons with amplicon size 100–250 pb. The PCR product was purified by ExoSAP-IT (USB Corporation, Cleveland, OH, USA) and directly sequenced using the dideoxy chain termination method in ABI Prism Big Dye^®^ Terminator v3.1 Cycle Sequencing Kit (Applied Biosystems, Waltham, MA, USA). Sequencing was processed in a 3500 Genetic Analyzer (Applied Biosystems, Waltham, MA, USA) and analyzed by means of the Sequencing Analyzer Software (Life Technologies, Carlsbad, CA, USA) comparing obtained results with the reference sequence from hg19.

## 5. Conclusions

Our results showed that rare intronic variants affecting the most canonical splice sites displayed in 100% of cases that they would affect the splicing product, possibly causing aberrant isoforms. However, 25% of these cases (1/4) showed normal splicing, contradicting the in silico results. On the contrary, in silico results predicted an effect in 0% of cases, and experimental results showed >20% (3/14) unpredicted aberrant splicing. Thus, deep intron variants are likely predicted to not have an effect, which, based on our results, might be an underestimation of their effect and, therefore, of their pathogenicity classification and family members’ follow-up.

## Figures and Tables

**Figure 1 ijms-23-12640-f001:**
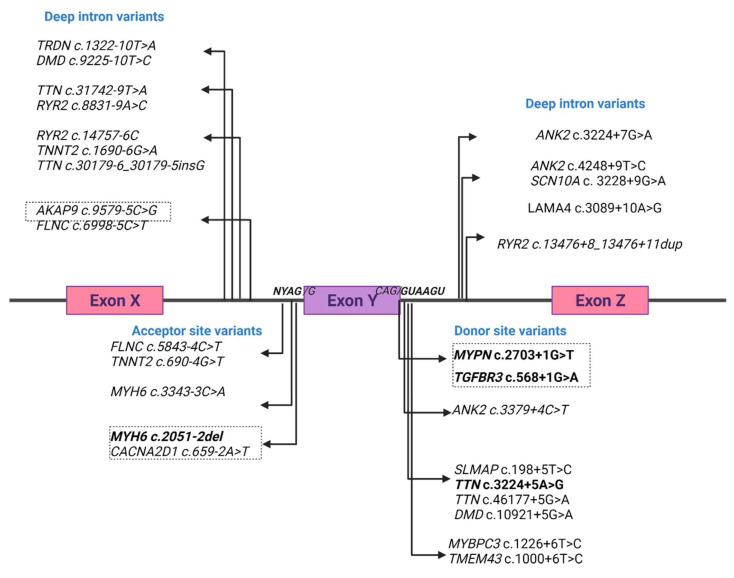
Schematic representation of selected intronic variants. Bold type indicates aberrant splicing shown experimentally. Dotted line indicates in silico predicted strong abnormal splicing.

**Figure 2 ijms-23-12640-f002:**
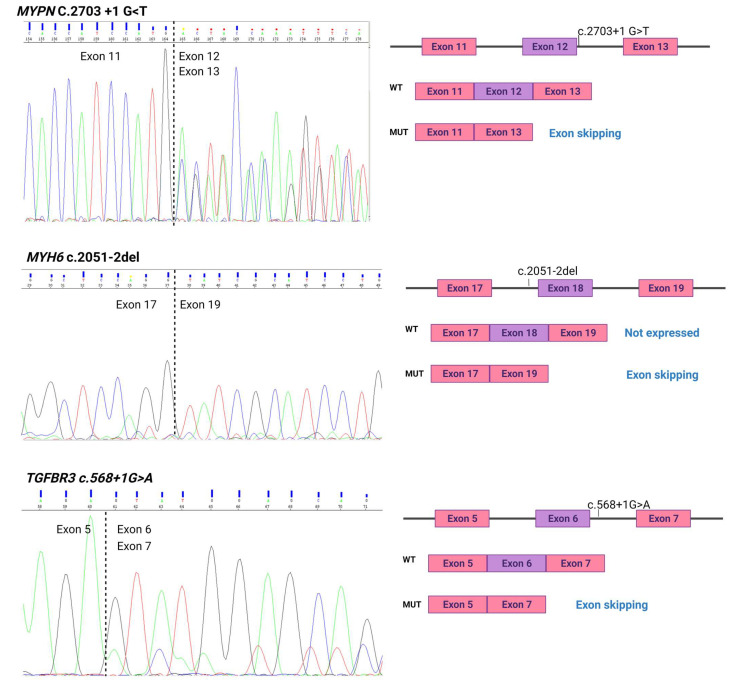
Aberrant splicing products identified for the most classical canonical variants.

**Figure 3 ijms-23-12640-f003:**
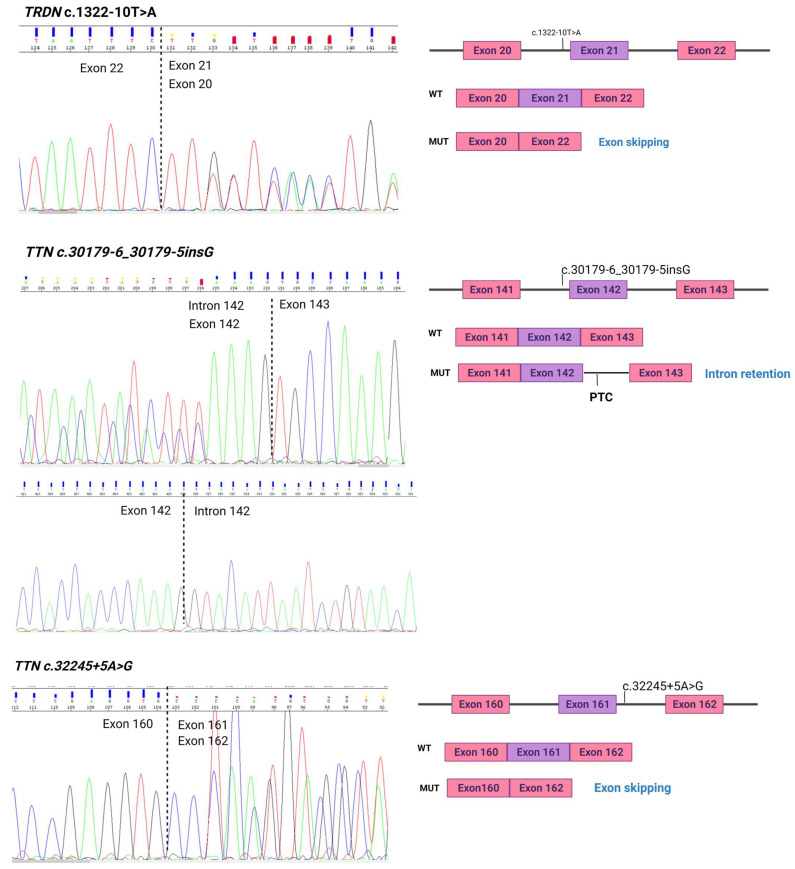
Unpredicted aberrant splicing products.

**Figure 4 ijms-23-12640-f004:**
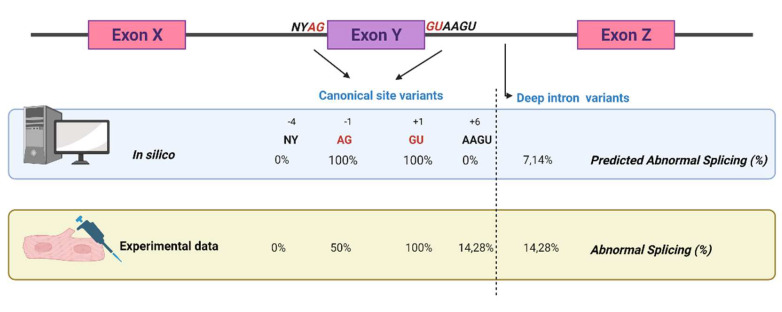
Representation of predicted abnormal splicing vs. experimentally shown. Canonical variants are represented based on their location (−3/4 and −1/2 for acceptor sites and +1/2 and +3/6 for donor sites). Deep intron variants include all variants beyond acceptor and donor consensus sites).

**Figure 5 ijms-23-12640-f005:**
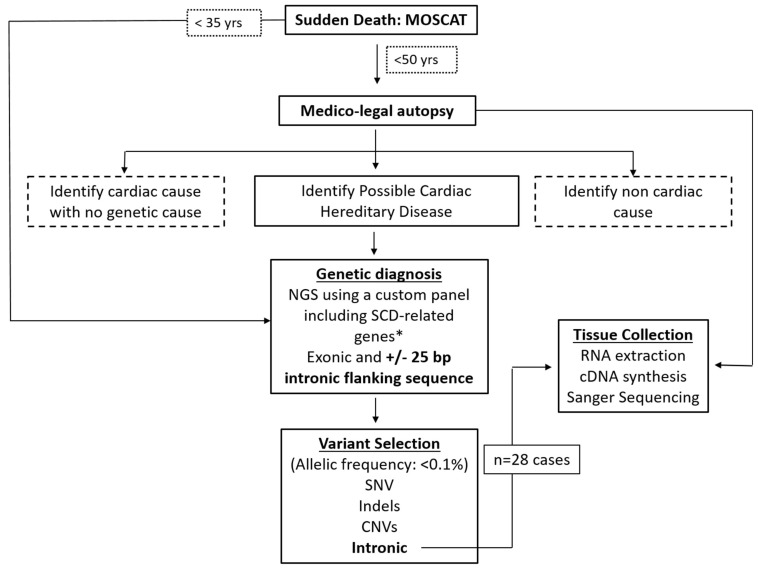
Sudden death: postmortem examination workflow of MOSCAT project. Dotted lines: excluded for the study. In individuals younger than 35 years old, toxicology screening, forensic examination and genetic analysis were performed simultaneously. * SCD-related genes included in panel (Appendix A).

**Table 1 ijms-23-12640-t001:** Autopsy details for SCD cases. F = female, M = male, SUD = sudden unexplained death, HCM = hypertrophic cardiomyopathy. Toxicology analysis showed detectable levels of drugs.

Case	Gender	Age	Autopsy Results	Site of Death	Histology	Toxicology Analysis	Intronic Variant	Other Variants
1	M	43	SUD	Home	Negative	Antiepileptic drugs	*ANK2* c.4248+9T>C	*MYBPC3* c.3295G>C (VUS), *MYH6* c.1492G>A (VUS), *MYH7* c.3277A>G (VUS), *DMD* c.8516C>T (VUS), *SLC22A5* c.728A>C (VUS)
2	F	36	SUD	Sleeping	Negative	Negative	*ANK2* c.3224+7G>A	*SLC8A1* c.11C>G (VUS), *SGCD* c.91C>T (VUS), *CTNNA3* c.2525G>A (VUS), *TTN* c.95312_95314del (VUS)
3	F	41	SUD	Sleeping	Negative	Cannabis	*ANK2* c.3379+4C>T	*DMD* c.7288G>C (VUS)
4	M	38	HCM	Prison	Positive	Codeine, cocaine, morphine	*LAMA4* c.3089+10A>G	*MYH7* c.166G>A (VUS), *MYBPC3* c.532G>A (VUS)
5	M	28	SUD	Sleeping	Positive	Caffeine, alcohol, cocaine, analgesics	*MYBPC3* c.1226+6T>C	None
6	M	24	SUD	Motorcycle racing	Negative	NA	*MYPN* c.2703+1G>T	*CACNA1C* c.6140C>T (VUS)
7	M	41	SUD	Drowning	Negative	NA	*RYR2* c.13476+8_13476+11dup	*KCNQ1* c.458C>T (VUS), *TTN* c.50521C>T (VUS), *TTN* c.25009A>G (VUS)
8	F	43	SUD	Home	Negative	Cocaine, cannabis	*SCN10A* c.3228+9G>A	None
9	M	35	SUD	Home	Negative	Negative	*TGFBR3* c.568+1G>A	*TTN* c.43913C>A (VUS), *LAMA4* c.4937A>G (VUS), *FBN2* c.8587C>T (VUS)
10	M	48	Possible myocardial infarction	Home	Positive	NA	*SLMAP* c.198+5T>C	Full gene duplication *TRDN* (VUS), *TTN* c.58483G>C (VUS)
11	M	10	SUD	Drowning	Negative	Negative	*TTN* c.32245+5A>G	*RYR2* c.10960G>A (VUS), *CALM1* c.307G>A (VUS), *RBM20* c.10G>A (VUS), *TTN* c.6958C>T (VUS)
12	M	32	SUD	Home	Negative	Caffeine, alcohol	*TTN* c.46177+5G>A	*SCN5A* c.1862del (LP), *MYH6* c.1002T>A (VUS)
13	F	18	Arrhythmogenic Cardiomyopathy	Hospital	Positive	Caffeine, anti-inflammatory	*MYH6* c.33433C>A	*CACNA1C* c.5474G>A (VUS), *TTN* c.92034A>C (VUS), *TTN* c.89827G>A (VUS)
14	M	44	SUD	NA	NA	NA	*AKAP9* c.95795C>G	*ABCC9* c.3522C>G (VUS), *LDB3* c.46C>T (VUS)
15	M	30	SUD	Home	NA	NA	*MYH6* c.20512del	*CASQ2* c.923C>T (VUS or LP in recessive CPVT context)
16	F	49	SUD	Hospital	Negative	Caffeine	*TMEM43* c.1000+6T>C	None
17	F	45	Cardiac ischemia	Home	Positive	NA	*DMD* c.10921+5G>A	None
18	M	23	SUD	Home	NA	Alcohol, cocaine, analgesics	*FLNC* c.58434C>T	*ABCC9* c.3979C>T (III o IV), *ACTN2* c.1370G>A (VUS)
19	M	44	SUD	Home	Positive	Negative	*TNNT2* c.6904G>T	*TRPM4* c.1652A>G (VUS); *SLMAP* c.1330G>A (VUS); *EYA4* c.347C>T (VUS)
20	M	26	Toxic	Home	Negative	Alcohol	*RYR2* c.147576C>T	None
21	F	19	SUD	Home	Negative	Caffeine, Alcohol	*CACNA2D1* c.6592A>T	*EN1* c.514G>T (VUS); *TNNT2* c.832C>T (LP)
22	M	44	SUD	Home	Negative	Alcohol, cocaine, cannabis	*TRDN* c.132210T>A	None
23	F	24	SUD	Public place	Negative	NA	*TTN* c.301796_301795insG	*NOTCH1* c.4013C>T (VUS); *MYH11* c.1151C>T (VUS); *KCNA5* c.460C>A (VUS)
24	M	27	SUD	Home	Negative	NA	*FLNC* c.69985C>T	*AKAP9* c.1204G>A (VUS); *PKP2* c.896G>T (VUS); *GAA* c.851C>G (VUS); *TTN* c.71381T>C (VUS); *TTN* c.41856A>T (VUS); *CACNA1* c.6388C>A (VUS)
25	M	41	Digestive	Home	Negative	Caffeine, alcohol, cocaine, analgesics	*RYR2* c.88319A>C	None
26	F	31	Cardiac ischemia	Public place	Positive	Negative	*TTN* c.317429T>A (TTN)	*GPD1L* c.372A>G (VUS);
27	F	35	SUD	Home	NA	Caffeine, cocaine	*TNNT2* c.6906G>A	*DSC2* c.857G>T (VUS); *TTN* c.59737A>C (VUS)
28	M	28	SUD	Public place	NA	Antiepileptic drugs	*DMD* c.922510T>C	*RYR2* c.2573C>T (VUS); *TTN* c.62666A>T (VUS); *TTN* c.47435T>C (VUS); *TTN* c.16066A>G (VUS)

**Table 2 ijms-23-12640-t002:** In silico prediction and experimental effect of variants. Allele frequency obtained from the gnomAD database (allele count/allele number of European population). NA = not available.

Case			Splicing Variant			
	Gene	Genetic Variant	Isoform	Allele Frequency	Predicted Effect	Functional Effect on mRNA Splicing
#1	*ANK2*	c.4248+9T>C	NM_001148.4	0/3476	Not affected	No aberrant splicing detected
#2	*ANK2*	c.3224+7G>A	NM_001148.4	4/129166	Not affected	No aberrant splicing detected
#3	*ANK2*	c.3379+4C>T	NM_001148.4	1/129086	Not affected	No aberrant splicing detected
#4	*LAMA4*	c.3089+10A>G	NM_001105206	NA	Not affected	No aberrant splicing detected
#5	*MYBPC3*	c.1226+6T>C	NM_000256.3	8/114656	Not affected	No aberrant splicing detected
#6	*MYPN*	c.2703+1G>T	NM_032578	NA	High	Exon 12 skipping
#7	*RYR2*	c.13476+8_13476+11dup	NM_001035.2	NA	Not affected	Intron retention PTC
#8	*SCN10A*	c.3228+9G>A	NM_006514.3	NA	Low/Moderate	No aberrant splicing detected
#9	*SLMAP*	c.198+5T>C	NM_007159	6/113652	Not affected	No aberrant splicing detected
#10	*TGFBR3*	c.568+1G>A	NM_003243	NA	High	Partial exon skipping
#11	*TTN*	c.32245+5A>G	NM_133378	NA	Not affected	No aberrant splicing detected
#12	*TTN*	c.46177+5G>A	NM_133378	NA	Not affected	No aberrant splicing detected
#13	*MYH6*	c.3343-3C>A	NM_002471	NA	Not affected	No aberrant splicing detected
#14	*AKAP9*	c.9579-5C>G	NM_005751.4	NA	Moderate	No aberrant splicing detected
#15	*MYH6*	c.20512del	NM_002471	NA	High	Exon skipping
#16	*TMEM43*	c.1000+6T>C	NM_024334.2	0/25062	Not affected	No aberrant splicing detected
#17	*DMD*	c.10921+5G>A	NM_004006	5/92407	Not affected	No aberrant splicing detected
#18	*FLNC*	c.58434C>T	NM_001458.4	NA	Not affected	No aberrant splicing detected
#19	*TNNT2*	c.6904G>T	NM_001001430.1	NA	Not affected	No aberrant splicing detected
#20	*RYR2*	c.147576C>T	NM_001035.2	NA	Not affected	No aberrant splicing detected
#21	*CACNA2D1*	c.6592A>T	NM_000722	NA	High	No aberrant splicing detected
#22	*TRDN*	c.1322-10T>A	NM_006073	NA	Not affected	Exon skipping
#23	*TTN*	c.301796_301795insG	NM_133378	NA	Not affected	No aberrant splicing detected
#24	*FLNC*	c.6998-5C>T	NM_001458.4	NA	Not affected	No aberrant splicing detected
#25	*RYR2*	c.8831-9A>C	NM_001035.2	NA	Not affected	No aberrant splicing detected
#26	*TTN*	c.31742-9T>A	NM_133378	NA	Not affected	No aberrant splicing detected
#27	*TNNT2*	c.690-6G>A	NM_001001430.1	NA	Not affected	No aberrant splicing detected
#28	*DMD*	c.9225-10T>C	NM_004006	NA	Not affected	No aberrant splicing detected

**Table 3 ijms-23-12640-t003:** Classification of splicing variants and the predicted effect. Underlined, the position is considered to be the most classical canonical site (+/−1 and 2).

Type of Mutation	Position	Effect on Splicing
Canonical acceptor site	1 to 4 bp 3′ NYAG/G sequence	Exon skipping (type I) Inclusion of the intron fragment or the removal of an exon fragment (type IV)
Canonical donor site	+1 to +6 bp 5′ CAG/GUAAGU sequence	Exon skipping (type I) Inclusion of the intron fragment or the removal of an exon fragment (type IV)
Deep intron	Within intron	Inclusion of an intron fragment (type II)
Exonic mutations affecting splicing	Within exon	Loss of exon fragment (type III)
Exonic mutations affecting splicing	Within exon	Exon skipping disruption of exonic splicing enhancers (type V)

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
