# Peer review of "Unpredicted Aberrant Splicing Products Identified in Postmortem Sudden Cardiac Death Samples"

_ijms, 2022, doi:10.3390/ijms232012640_

Round 1

Reviewer 1 Report (New Reviewer)

The current manuscript by Mònica Coll et al. investigated the correlation between aberrant splicing products to sudden cardiac death. The manuscript utilized many human cardiac tissues making the study translational. However, my major concern is the biological significance of the current study. Although there is some correlation between the intronic variants and sudden cardiac death, there is no data explaining how these variances lead to cardiac diseases. Some other concerns are discussed below. 

There are many experimental details lacking, such as how the cardiac tissues were processed to single cells and how genome DNA and RNA were extracted. Using what enzyme and at what temperature? 

How long the cardiac tissue was stored before processing for subsequent analysis? 

All of the Figures provided are low resolution and can not be interpreted. High-resolution images must be provided. 

It would be nice to use immunohistochemistry or immunofluorescence methods to detect the relevant pathways that were predicted to changes associated with the gene variances. 

Author Response

The current manuscript by Mònica Coll et al. investigated the correlation between aberrant splicing products to sudden cardiac death. The manuscript utilized many human cardiac tissues making the study translational. However, my major concern is the biological significance of the current study. Although there is some correlation between the intronic variants and sudden cardiac death, there is no data explaining how these variances lead to cardiac diseases. Some other concerns are discussed below. 

  • There are many experimental details lacking, such as how the cardiac tissues were processed to single cells and how genome DNA and RNA were extracted. Using what enzyme and at what temperature? 

We have included and detailed specifically the DNA extraction form post-mortem blood samples and the RNA extraction from frozen cardiac tissue:

DNA extraction from blood samples Genomic DNA was extracted from postmortem whole blood with the automated instrument Chemag-ic MSM I (PerkinElmer, Waltham, MA, USA). Spectrophotometric measurements were performed to assess quality ratios of absorbance (260/280:260/230 minimum of 1.8:2.2). DNA concentration was de-termined by Qubit fluorimeter (Thermo Fisher Scientific, Waltham, MA, USA).

RNA extraction from cardiac tissue. 25 mg of cardiac tissue were processed for each samples to extract RNA. Tissues were disrupted by rapid agitation in the presence of beads and lysis buffer using Tissuelyser LT (Qiagen) during 2 min at 30 Hz. Disruption and simultaneous homogenization occur by the shearing and crushing action of the beads as they collide with the cells. RNA extraction was performed using RNeasy Mini Kit (74106, Qiagen) and according to the manufacturer’s instructions This protocol includes an incubation of Proteinase K at 55 C during 10 min.

We changed the manuscript to include a more detailed information on DNA extraction (lines 359-365) and RNA extraction. (lines 384-389)

  • How long the cardiac tissue was stored before processing for subsequent analysis? 

All 28 cases suffered SCD in the period 2016-2021. DNA extraction from blood post mortem samples were processed immediately and DNA were stored at -20ºC. Cardiac samples were stored at -80ºC from their initial recollection until their processing (January-March 2022), with no prior freeze-thaw cycles.

Added information (lines 384-385)

  • All of the Figures provided are low resolution and cannot be interpreted. High-resolution images must be provided. 

 We have included high-resolution images in this new submission.

  • It would be nice to use immunohistochemistry or immunofluorescence methods to detect the relevant pathways that were predicted to changes associated with the gene variances. 

We agree that it would be interesting to study which are the relevant pathways predicted to be affected because of gene variances. At this stage it is no feasible to perform an immunolabeling study with the samples included in this study and, it is beyond the scope of our study which is mainly focused on determining the effect of intronic variants in the final mature transcript to contribute in the current knowledge of these variants in terms of future pathogenicity prediction.

On the other hand, we sent the manuscript to the MDPI English editing for professional proofreading and it was extensively revised

Reviewer 2 Report (New Reviewer)

The manuscript “Unpredicted aberrant splicing products identified in postmortem sudden cardiac death samples” by Coll et al., investigated rare intronic variants in 28 heart samples using a targeted custom panel including 85 SCD genes. The authors discussed in the text already cohort sample size is too limited to represent their findings. Furthermore, the authors missed many important citations in the text. I would recommend performing whole genome and RNA-seq to discover more variants as current methods in this field. I am sorry that I can not support the publication of your manuscript at this stage of your findings and wish you all the best with your further study. Thank you.

Author Response

The manuscript “Unpredicted aberrant splicing products identified in postmortem sudden cardiac death samples” by Coll et al., investigated rare intronic variants in 28 heart samples using a targeted custom panel including 85 SCD genes. The authors discussed in the text already cohort sample size is too limited to represent their findings. Furthermore, the authors missed many important citations in the text. I would recommend performing whole genome and RNA-seq to discover more variants as current methods in this field. I am sorry that I cannot support the publication of your manuscript at this stage of your findings and wish you all the best with your further study. Thank you.

This study is part of a previous study MOSCAT were all cases of sudden death occurred in Catalonia during 10 years (2010-2020) with legal autopsy were genetically analysed. As for the nature of the MOSCAT project, only heart sample recollected during the medico-legal autopsy suspected from SCD were send to be genetically analysed in our laboratory. The primary objective is genetically identify putative causal mutations in the SCD genes (using a 85 SCD genes custom panel) since molecular screening for pathogenic mutations in SCD related genes is the common practice in the molecular diagnosis  according to current guidelines (Wilde et al. 2022).  We agree with the reviewer that some citations were missed and we included them in the new manuscript version.

We know the sample size is limited, however, obtaining fresh well conserved heart from SCD cases in a forensic protocol together with a positive genetic results (rare intronic variant in a SCD related gene) is very challenging. In this sense, this 28 cases were selected from an initial study of more than 800 cases of SCD and only this 28 cases carried rare intronic variants and fulfilled inclusion criteria for being considered SCD.

We agree that a combined approach of whole genome and RNA-seq would allow as to discover more variants. However, at this stage, it is unrealistic for us to perform whole genome and RNA-seq due to economic and technical limitations.

On the other hand, we sent the manuscript to the MDPI English editing for professional proofreading and it was extensively revised

Reviewer 3 Report (New Reviewer)

In this paper, Coll and colleagues analyzed heart samples from 28 sudden cardiac death (SCD) cases using 85 SCD genes custom panel sequencing and splicing products sequencing. They identified a number of variants around the target genes as well as unpredicted aberrant splicing events. The study is of potential value to address the genetic etiology of SCD. A major deficiency is that the cohort size is too small to support the conclusion, and the potential molecular mechanisms have never been examined or discussed. Besides, references for many claims in the manuscript are missing. The scientific English is likewise inadequate; therefore improvement is needed.

Specific comments:

1.       The author stated many times on that “molecular screening for pathogenic mutations in SCD related genes is common practice”, please add the reference literature.

2.       Line 64-66, please add the reference literature.

3.       Line 124-125, please add the reference literature.

4.       Line 128, line 137, please add the reference literature

5.       The author used 85 SCD genes custom panel sequencing, and please provide related references.

6.       Please indicate the full name of SCD in the abstract.

7.       Please use a period (full stop) to show decimal point. For example,

8.       Line 95, typo error of “a range of 1849 years old”

9.       Line 124, typo error, “considereed” should be “considered”

10.   Line 136, typo error, “undetecTable” should be “undetectable”

Author Response

In this paper, Coll and colleagues analyzed heart samples from 28 sudden cardiac death (SCD) cases using 85 SCD genes custom panel sequencing and splicing products sequencing. They identified a number of variants around the target genes as well as unpredicted aberrant splicing events. The study is of potential value to address the genetic etiology of SCD. A major deficiency is that the cohort size is too small to support the conclusion, and the potential molecular mechanisms have never been examined or discussed. Besides, references for many claims in the manuscript are missing. The scientific English is likewise inadequate; therefore, improvement is needed.

We agree with the reviewer about the small cohort size. For this reason, we emphasise that in the limitations sections.  These 28 cases we selected from an initial study of more than 800 cases of SCD and only this 28 cases carried rare intronic variants and fulfilled inclusion criteria for being considered SCD. The initial 800 samples were collected during the last 5 years. SCD cases occurred in Catalonia suffered by individuals < 50 years- old which required a medico-legal autopsy and suspected from SCD, were send to be genetically analysed in our laboratory. We know the sample size is limited, however, obtaining fresh well conserved heart from SCD cases in a forensic protocol together with a positive genetic results (rare intronic variant in a SCD related gene) is challenging. For this reason, we believe this results may still be valuable despite the cohort size.

We agree with the reviewer that important citations were missed and we made changes in the manuscript to improve that.

The scientific English is likewise inadequate; therefore, improvement is needed.

We sent the manuscript to the MDPI English editing for professional proofreading and it was extensively revised.

Specific comments:

  1. The author stated many times on that “molecular screening for pathogenic mutations in SCD related genes is common practice”, please add the reference literature. Added
  2. Line 64-66, please add the reference literature. Added
  3. Line 124-125, please add the reference literature. Added
  4. Line 128, line 137, please add the reference literature. Added
  5. The author used 85 SCD genes custom panel sequencing, and please provide related references. A supplemental table with the comprised genes has been included. There was not references available due to was custom panel designed by our own research group. This custom panel was based on the recommended genes by the European Heart Rhythm Association (EHRA)/ Heart Rhythm Society (HRS) and also candidate genes by published bibliography.
  6. Please indicate the full name of SCD in the abstract. Done
  7. Please use a period (full stop) to show decimal point. For example,
  8. Line 95, typo error of “a range of 1849 years old”. Done
  9. Line 124, typo error, “considereed” should be “considered” done
  10. Line 136, typo error, “undetecTable” should be “undetectable” done

Round 2

Reviewer 1 Report (New Reviewer)

I don't have further comments

Author Response

I don't have further comments

Thanks for all the reviewer' comments during the revision process. We  have submitted  the final version of the manuscript

Reviewer 2 Report (New Reviewer)

I accepted the author's explanation for the limited sample size and methods in the current situation. The revised version is improved overall. I have just one suggestion about table1. Authors should make realignment of each row and adjust the space to make it easy to read. Other than that, I don’t have any further comments. Thank you.

Author Response

I accepted the author's explanation for the limited sample size and methods in the current situation. The revised version is improved overall. I have just one suggestion about table1. Authors should make realignment of each row and adjust the space to make it easy to read. Other than that, I don’t have any further comments. Thank you.

We made the suggested changes on table 1 to make it easy to read.

Reviewer 3 Report (New Reviewer)

The authors have addressed my questions. 

Author Response

The authors have addressed my questions. 

Thanks for all the reviewer' comments during the revision process. We  have submitted  the final version of the manuscript

This manuscript is a resubmission of an earlier submission. The following is a list of the peer review reports and author responses from that submission.

Round 1

Reviewer 1 Report

Major comments:

My major criticism is about the study design and data interpretation. First, the sample size is too limited to provide a solid conclusion, especially when the donors of the samples have very diverse backgrounds (age, toxicology analysis, etc). Second, a defect of this design, from my opinion of view, is a lack of control. I wonder whether the results might look similar if we analyzed another group of 28 non-SCD samples. Third, the authors have explicitly listed all detected splice site variants in each of these 28 samples. However, not a single type of splicing variant is reported with noticeable prevalence. Fourth, the concordance between DNA variant and RNA splicing cannot be studied using the approach of the authors. The allele frequency of these variants are not 100%, meaning that a variant of splice site at only 5% might have very little impact on the RNA transcripts. The authors should probably only experimentally validate the RNA alterations predicted from those DNA variants with very high allele frequency in the samples. I tried to search for such information (allele frequency) from the manuscript but failed to find a clue. Overall, I don’t think the approaches used in this study are sufficient to discover the splice site variants of SCD. 

Other minor comments:

1) The authors have not provided a clear explanation of ‘experimental validation’. I was able to find cDNA synthesis method from the Method section but I failed to find how the RNA splicing was analyzed.

2) The authors could improve a little bit on the glossary of specific terms. For example, the first appearance of ‘LP’ is in line 90, section 2.1. However, the explanation of LP, which is “likely pathogenic” by the authors’ definition, is in line 354 (materials and methods section). This makes it really hard to read this manuscript.

3) I would recommend the authors use some proofreading to improve the grammar of English.

4) The title for Table 1 is on page3 but the table itself is on page4, with figure 1 in between. The reading experience becomes a bit hard.

Reviewer 2 Report

In this article, authors evaluate splicing abnormalities in 28 cases of sudden cardiac death cases.

The study has some limitations, and it is not well discussed and explained.

The text has to be extensively revised, and methods are poorly described.

It is not clear how the samples are collected and processed. The investigation steps need to be deeply described.

Moreover, the limited number of patients do not allow a general comprehension of the biological significance of the reported data.

Reviewer 3 Report

ijms-1866276

Molecular characterization of splice site variants in sudden cardiac death cases

Mònica Coll, Anna Fernandez-Falgueras, Anna Iglesias, Bernat del Olmo, Laia Nogué-Navarro, Adrià Simon, Alexandra Pérez Serra, Marta Puigmulé, Laura López, Ferran Picó, Mònica Corona, Marta Vallverdú-Prats, Coloma Tiron Oscar Campuzano Josep Castella, Ramon Brugada, Mireia Alcalde

The authors claim this manuscript tried to elucidate the functional effect of potential splice site variants in 28 cases of SCD by performing molecular studies to assess splicing abnormalities in myocardium postmortem human tissue and evaluate possible discrepancies between computational predictions and experimental evidence. They think their results showed that intronic variants affecting canonical splice sites showed very different in silico predictions depending on their location. In silico predictions for the most classical canonical splice variants (/+ 1 and 2) are prone to predict as possibly affecting splicing products. In silico predictions revealed to have null sensibility to be interpret rigorously the effect of variants on the splicing product. Experimental studies showed that splicing aberrations may be much more complex than their location in the genetic sequence.

Many splice site mutations and deep intronic mutations were already reported with numerous numbers of disease-causing genes. Mutations at 5’ splice site (+1 or 2) are already known and predictable to cause exon skipping. Although the authors mention ‘molecular characterization’, only RT-PCR followed by sequencing was carried out. No other novel information about aberrant splicing was presented.

It seems to me that this manuscript does not contain any novel findings for both splicing and disease field. I have to conclude this manuscript cannot be accepted by IJMS journal.